# Immunopathogenesis of Immune Checkpoint Inhibitor Induced Myocarditis: Insights from Experimental Models and Treatment Implications

**DOI:** 10.3390/biomedicines11010107

**Published:** 2023-01-01

**Authors:** Chun-Ka Wong, Tsun-Ho Lam, Song-Yan Liao, Yee-Man Lau, Hung-Fat Tse, Benjamin Y. F. So

**Affiliations:** 1Department of Medicine, School of Clinical Medicine, Li Ka Shing Faculty of Medicine, The University of Hong Kong, Hong Kong SAR, China; 2Cardiac and Vascular Center, The University of Hong Kong-Shenzhen Hospital, Shenzhen 518053, China; 3Hong Kong-Guangdong Stem Cell and Regenerative Medicine Research Centre, The University of Hong Kong and Guangzhou Institutes of Biomedicine and Health, Hong Kong SAR, China; 4Centre for Stem Cell Translational Biology, Hong Kong SAR, China

**Keywords:** autoimmunity, cancer, immune checkpoint inhibitor, immunotherapy, leukocyte, myocarditis, transgenic mice

## Abstract

Despite the extraordinary success of immune checkpoint inhibitors (ICIs) in cancer treatment, their use is associated with a high incidence of immune-related adverse events (IRAEs), resulting from therapy-related autoimmunity against various target organs. ICI-induced myocarditis is one of the most severe forms of IRAE, which is associated with risk of hemodynamic compromise and mortality. Despite increasing recognition and prompt treatment by clinicians, there remain significant gaps in knowledge regarding the pathophysiology, diagnosis and treatment of ICI-induced myocarditis. As the newly emerged disease entity is relatively rare, it is challenging for researchers to perform studies involving patients at scale. Alternatively, mouse models have been developed to facilitate research understanding of the pathogenesis of ICI-induced myocarditis and drug discovery. Transgenic mice with immune checkpoint genes knocked out allow induction of myocarditis in a highly reproducible manner. On the other hand, it has not been possible to induce ICI-induced myocarditis in wild type mice by injecting ICIs monotherapy alone. Additional interventions such as combinational ICI, tumor inoculation, cardiac sarcomere immunization, or cardiac irradiation are necessary to mimic the underlying pathophysiology in human cancer patients and to induce ICI-induced myocarditis successfully. This review focuses on the immunopathogenesis of ICI-induced myocarditis, drawing insights from human studies and animal models, and discusses the potential implications for treatment.

## 1. Introduction

### 1.1. A New Era of Cancer Immunotherapy

The introduction of immune checkpoint inhibitors (ICIs) has revolutionized the modern-day treatment of cancer. Since the approval of the anti-cytotoxic T-lymphocyte-associated protein 4 (CTLA4) antibody ipilimumab in 2011 by the US Food and Drug Administration [1], followed a few years later by the anti-programmed death 1 (PD-1) antibodies nivolumab and pembrolizumab [2,3], nine different ICIs have now been approved for a variety of solid tumors. They have been extensively applied as monotherapy or in combination with chemotherapy, in the neoadjuvant, adjuvant or palliative settings [4].

Modern T cell-directed immunotherapy is highly effective, associated with high response rates and, crucially, capable of achieving long-lasting remission even in the context of advanced metastatic disease, raising the possibility of a cure for cancer.

### 1.2. Mechanisms of Immune Checkpoint Inhibitors

ICIs enhance the immune response against cancer cells and achieve anti-neoplastic effects [5] (Figure 1). Cancer cells can escape immunosurveillance by changing their surface antigens to evade detection and destruction by lymphocytes, including the expression of ligands that can inhibit T cell responses downstream. These ligands are known as immune checkpoints and normally function to prevent the development of autoimmunity. Two of the most well-studied checkpoints that may be targeted for the treatment of cancer are the CTLA4 and the PD-1/PD-L1 (programmed death ligand-1) pathways [6,7,8,9].

In lymph nodes draining tumors, antigen-presenting cells normally present tumor-associated peptides to T cells [10]. Activated T cells differentiate to provide help to facilitate B cell maturation, resulting in a robust and antigen-specific immune response [11]. Classically, T cell activation occurs through three signals. First, antigen presentation occurs via interaction between the T-cell receptor and antigen-presenting cells, which then activates downstream pathways including the well-studied calcineurin pathway [12]. Second, co-stimulatory signals are required for T cell activation, without which T cells would rapidly become anergic. One of the most well-studied co-stimulatory receptors is CD28, which binds to CD80 and CD86. A homologue of the CD28 molecule known as CTLA4 is induced on T cells and acts as a negative regulator of T cell activity by competitively binding CD80 and CD86 [13]. Third, cytokines provide additional signaling that facilitates T cell activation and development of a mature T cell repertoire [14]. Some of the better studied cytokines include IL-12 and type 1 interferons. Although it remains controversial whether all three signals are required for T cell activation in all situations, modulation of these pathways, individually or in concert, have been successfully used to alter the immune response in various diseases. In the context of cancer immunotherapy, antibodies directed against the inhibitory co-stimulatory molecule CTLA4, such as ipilimumab, are used to augment the immune response against tumor-associated peptides [1,15,16,17].

Meanwhile, the PD-1 pathway plays a role in the regulation of central as well as peripheral tolerance of T cells, by the binding of PD-1 to the PD-L1 ligand. PD-1 plays a role in selecting out autoreactive T cells in the thymus, and it also functions in secondary lymphoid organs such as lymph nodes to regulate T cell activation through diverse mechanisms [18]. Although many of these mechanisms have yet to be elaborated, interaction with the T cell receptor and modulation of cytokine expression have been demonstrated. These mechanisms appear to be distinct from those regulated by the CTLA4 co-stimulatory pathway. Furthermore, endothelial expression of PD-1 functions to prevent migration of T cells to non-lymphoid tissues [19]. Finally, in peripheral organs, PD-1 expression prevents infiltrating T cells from acquiring effector functions, preventing autoimmune attack on target organs [20]. Cancer cells, as well as other cells in the tumor microenvironment, appear to be enriched in PD-1 and PD-L1 expression, resulting in anergy of T cells directed against tumor antigens in organs, including in metastatic tumor deposits [21]. Therefore, antibodies directed against PD-1, such as pembrolizumab and nivolumab, or PD-L1, such as atezolizumab and durvalumab, have been used to restore T cell-mediated immunosurveillance against proliferating cancer cells [3,22,23,24,25,26,27,28,29,30].

Other immune checkpoints have also been investigated for potential application in oncology, including lymphocyte activation gene-3 (LAG-3) [31]. LAG-3 is highly homologous with CD4, which is expressed on all activated T cells, and may be found on activated T cells, dendritic cells, natural killer cells as well as certain B cells. Due to its high homology with CD4, it is postulated to compete with CD4 binding to negatively regulate T cell activation [31]. As an illustration, the binding affinity of major histocompatibility complex class II (MHC Class II) for LAG-3 is 100 times that for CD4, allowing LAG-3 to play a critical role in dampening the T cell response in secondary lymphoid organs upon antigen presentation [32]. LAG-3 has also been demonstrated in tumor-infiltrating lymphocytes in a variety of solid tumors [33]. Given its likely role in the immunopathogenesis of cancer, the anti-LAG-3 antibody relatlimab has also recently received approval for metastatic melanoma, and clinical trials investigating relatlimab in conjunction with other ICIs in various cancers are currently underway [34].

### 1.3. Immune-Related Adverse Events

#### 1.3.1. Diverse Presentations of Immune-Related Adverse Events

Despite the extraordinary success of ICIs in cancer treatment, their use is associated with a high incidence of immune-related adverse events (IRAEs). IRAEs result from therapy-related autoimmunity against various target organa. The skin, liver, gastrointestinal tract, kidneys and endocrine organs, including the pituitary, thyroid and adrenal glands, have been most extensively described [35]. Although over one third of all patients treated with conventional ICIs manifest at least one IRAE [36], the pattern of organ involvement differs significantly, likely as a result of the interplay between underlying autoimmune diatheses, other medical comorbidities, intercurrent acute illnesses, environmental antigen and drug exposures and the type and dose of ICI administered [37,38,39]. Interestingly, tumor response seems to be better in patients with IRAEs than those without, suggesting a trade-off between treatment efficacy and side effects [40]. IRAEs may present with varying degrees of organ dysfunction, some of which may be organ- or even life-threatening. (Table 1) Active surveillance and a high index of suspicion are required to detect IRAEs early, and treatment with corticosteroids with or without other immunosuppressants may be required in a significant proportion of cases.

#### 1.3.2. Epidemiology, Presentation, Diagnosis and Contemporary Treatment of Immune Checkpoint Inhibitor-Induced Myocarditis

Of the IRAEs, one of the most feared is ICI-induced myocarditis. Fulminant cases of myocarditis after ICI therapy were first reported in 2016, and multiple case series from around the world have followed since [62,63,64,65,66,67]. Although myocarditis associated with ICIs occurs rarely, especially when compared with other IRAEs, it is associated with high mortality rates and significant morbidity even in survivors. Studies report an incidence rate ranging from 0.27% to 1.14% [62,63], but this may be a gross underestimation of the actual incidence, given inconsistent reporting practices and frequent underrecognition and underdiagnosis. Pharmacovigilance studies suggest a five-fold increased risk of developing myocarditis among recipients of ICIs, with an even higher risk among those receiving combinations of ICIs [68].

Similar to myocarditis caused by other etiologies, ICI-induced myocarditis patients may experience heart failure, arrhythmias, pericarditis and pericardial effusion, and sudden cardiac arrest [42]. Nevertheless, certain symptoms and associations appear to be unique to this newly defined clinical entity. Arrhythmias, including atrial fibrillation, ventricular arrhythmias and conduction abnormalities, appear to be common in ICI-induced myocarditis [42]. Although systolic dysfunction resulting in heart failure with cardiogenic shock is observed in fulminant cases of myocarditis, smoldering cases with raised cardiac biomarkers (such as elevated cardiac troponin) and preserved systolic function have also been described, reflecting the broad spectrum of severity of ICI-induced myocarditis, and suggesting that many less severe cases may be easily missed by clinicians [62,69]. Importantly, many case series have highlighted a correlation between fulminant myocarditis and skeletal myositis and myasthenia-like “3M Syndrome”, although the exact mechanisms for these associated extracardiac manifestations remain speculative [67,70,71,72]. The presence of any one of the triad should prompt close monitoring and evaluation to identify the other diagnoses and guide the use of immunomodulatory therapy.

There are few guidelines regarding how best to screen for or diagnose ICI-induced myocarditis [73,74]. Although a baseline electrocardiogram and possibly an echocardiogram are probably indicated in most cancer patients, screening using cardiac biomarkers such as cardiac troponin have yet to be validated in cancer populations, including those on ICIs. In patients clinically suspected to have myocarditis, the gold standard for diagnosis remains an endomyocardial biopsy [75]. However, endomyocardial biopsy may often be difficult and high-risk to perform in cancer patients, and less invasive modalities of investigation such as cardiac magnetic resonance imaging (MRI) or serum biomarkers have been suggested as alternatives, but these modalities have not been thoroughly validated for ICI-induced myocarditis.

There is equally little high-quality evidence to guide the treatment of ICI-induced myocarditis. Like with other severe IRAEs, high-dose corticosteroids are typically advocated for myocarditis, including the use of pulsed methylprednisolone at doses of up to 1 g per day [73,74]. However, ICI-induced myocarditis is often refractory to steroid monotherapy, and ongoing myocardial damage with elevated cardiac troponin is typically observed for months [67]. Various other immunomodulatory treatments have been tried in ICI-induced myocarditis and reported in small case series, often with modest degrees of success. Second-line agents that have been reported include mycophenolate mofetil, [76,77] calcineurin inhibitors [78], JAK inhibitors [79], anti-TNFα [80], anti-CD20 [81] and anti-T cell antibodies such as the polyclonal antibody anti-thymocyte globulin [80] anti-IL2 monoclonal antibodies such as basiliximab [67] and the CTLA4 costimulation modulator abatacept [67,76]. Plasmapheresis [80,82] and intravenous immunoglobulins [80] have also been employed to remove as-yet-unidentified pathogenic antibodies. Large randomized controlled trials have not been and are unlikely to be performed for ICI-induced myocarditis, and contemporary treatment of ICI-induce myocarditis is currently based on experiences reported in case series, with their inherent risk of reporting bias, as well as expert opinion and extrapolation from treatment of other IRAEs and other causes of myocarditis. Despite multimodal immunomodulatory treatments, however, the mortality of ICI-induced myocarditis remains unacceptably high.

Clearly, despite increasing recognition and prompt treatment by clinicians, there remain significant gaps in knowledge regarding the pathophysiology, diagnosis and treatment of ICI-induced myocarditis. This review will focus on the immunopathogenesis of ICI-induced myocarditis, drawing insights from human studies and animal models, and explore potential therapeutic targets and pharmacological options identified from these studies.

## 2. Human Studies of ICI-Induced Myocarditis

Prior to the era of ICIs, researchers have extensively investigated the pathophysiology of myocarditis. Pathogens including viruses, bacteria, parasites and fungi may cause myocarditis both by direct cytopathic effects on cardiomyocytes and by triggering an autoimmune reaction against the myocardium [83,84,85,86,87]. Autoimmune myocarditis may also occur in the presence of systemic autoimmune illnesses such as eosinophilic granulomatosis with polyangiitis (EGPA), dermatomyositis or sarcoidosis. In real-world clinical practice, obvious triggers for isolated myocarditis are often absent, though it has been speculated that autoimmunity plays a key role in the pathogenesis as autoantibodies against the myocardium are commonly isolated from the sera of patients with myocarditis [88]. Pathogenic mutations in genes such as desmoplakin may cause recurrent myocarditis and cardiomyopathy [89]. Drugs including antipsychotics, salicylates, cytotoxic agents, ICIs and vaccines are known to cause myocarditis by inducing allergic responses or via other mechanisms such as transient suppression of anti-inflammatory cytokines [90,91]. As ICI-induced myocarditis is a newly defined entity, its pathogenic mechanism remains poorly understood. Table 2 summarizes the current understanding of the pathogenesis of ICI-induced myocarditis derived from human studies.

### 2.1. Endomyocardial Histology, Immunohistochemistry and Tissue Transcriptomics

As mentioned above, an endomyocardial biopsy is the gold standard for diagnosis of myocarditis. However, differentiation of ICI-induced myocarditis from other pathological causes of myocarditis is difficult with light microscopy alone. Immunophenotyping techniques have been employed to delineate the profile of immunological activation in ICI-induced myocarditis [92,93,94,95]. Most of the studies published to date are case series without case-control comparisons. The data provided by these reports are nevertheless invaluable owing to the rarity of the disease entity. Data from case series of endomycardial biopsies from affected patients suggest that ICI-induced myocarditis is typically characterized by a lymphocytic infiltrate in the myocardium dominated by CD8+ cytotoxic T cells, although there may also be a significant proportion of CD68+ or CD163+ macrophages and CD4+ helper T cells [92,93,94,95]. Although expression of granzyme B by CD8+ cytotoxic T cells in tissues suggests a direct pathogenic role in the development of tissue damage, the ratio of CD8+ cytotoxic T cells to total number of T cells in the tissue section (defined by CD3 positivity on immunohistochemistry) was found to be similar in high-grade and low-grade ICI-induced myocarditis, suggesting that other factors may modulate the severity of disease [95]. Interestingly, compared to grade 2R acute cellular rejection of heart transplant recipients, the ratio of CD68+ macrophages to CD3+ T cells in the myocardium was higher in ICI-induced myocarditis, suggesting a prominent role played by infiltrating macrophages, contributing to a more lymphohistiocytic pathology [92]. However, the myocardial CD68+/CD3+ ratio was found to be similar in high-grade and low-grade ICI-induced myocarditis [92]. Eosinophils are infrequently seen, mostly in patients with high-grade myocarditis, in contrast with eosinophilic myocarditis typically associated with hypereosinophilic syndromes [92,94,95]. It remains controversial whether PD-1 or PD-L1 positivity is a defining feature of ICI-induced myocarditis [95]. Some preliminary reports suggest that the level of PD-L1 expression by CD68+ macrophages may correlate with the severity of myocarditis [92].

Different gradings of myocarditis based on microscopic findings have been proposed in the literature, but these are largely extrapolated from other causes of myocarditis and do not necessarily have either diagnostic specificity or prognostic significance for ICI-induced myocarditis. Higher grades of myocarditis additionally show significant cardiomyocyte injury on top of an inflammatory infiltrate, which may be better appreciated on electron microscopy. Typical findings include cardiomyocyte necrosis with myofibrillar degeneration and sarcoplasmic tubular dilatation, although these findings are not specific for ICI-induced myocarditis [94]. Some studies show that a proportion of cardiomyocytes may be PD-L1+, particularly at sites of cardiomyocyte injury from patients with high-grade ICI-induced myocarditis [92,94,95]. Further studies may be needed to clarify if these are specific features for ICI-induced myocarditis.

Furthermore, immunohistochemical studies of endomyocardial biopsy specimens have shown that necrotic cardiomyocytes are frequently positive for the complement activation product C4d in ICI-induced myocarditis, suggesting activation of either the classical or mannose-binding lectin (MBL) pathway of complement. This may point to a role for an antigen-antibody interaction and immune complex formation with complement fixation [92]. It is not clear whether these immune complexes are the result of interactions between circulating antibodies and in situ myocardial antigens, or were preformed and subsequently deposited in the myocardium. Nevertheless, to date, no study has demonstrated the presence of autoantibodies against cardiomyocytes in the serum of patients with ICI-induced myocarditis.

Tissue transcriptomics have rarely been reported to date. In one study, RNA sequencing was performed to compare ICI-induced myocarditis with viral myocarditis and dilated cardiomyopathy. ICI-induced myocarditis was associated with differential expression of genes that involved multiple inflammatory pathways, especially interferon responses [96]. These findings correlate well with the profile of inflammatory cells observed in ICI-induced myocarditis, as interferons are secreted predominantly by helper T cells, cytotoxic T cells and macrophages among other inflammatory cells.

### 2.2. Peripheral Blood Mononuclear Cell Profile

The onset of ICI-induced myocarditis is mirrored by perturbations in the repertoire of peripheral blood mononuclear cells (PBMCs). In case series, the absolute lymphocyte count was significantly reduced, with an elevated neutrophil-to-lymphocyte ratio [97]. The T cell phenotype among patients with ICI-induced myocarditis was characterized by a T_H_1 type response, with a reduction in T_H_2 cells and increase in T_H_1, T_H_17 and regulatory T cells. CXCR3 expression was increased among CD8+ memory T cells, a pattern which is associated with generation of effector memory T cells. CXCR3 expression was also increased in memory B cells, which may be associated with co-expression of the IgG1 subclass [98]. However, many of the above studies were uncontrolled case series. It is difficult to determine whether these changes in the repertoire of PBMCs may simply be reactive to underlying malignancy or to concurrent chemotherapy and/or radiotherapy, and no definite conclusions can be drawn. Importantly, a more salient comparison with patients given ICIs but who did not develop myocarditis showed that patients who developed myocarditis had a smaller proportion of naïve CD8 cells and larger proportion of terminally differentiated effector memory CD45RA re-expressing CD8+ T cells (Temra). T-cell receptor sequencing determined that these Temra CD8+ cells were of clonal origin, and their expression was associated with elevated levels of proinflammatory and cardiotropic chemokines including CCL5, CCL4 and CCL4L2. They also had higher levels of cytotoxic and activation markers’ expression, including granzyme B and interleukin-32 when compared to other CD8+ T cells [99]. The macrophage/monocyte expression pattern was also different, with an increased expression of “non-classical” monocytes more typically associated with an inflammatory response and capacity for antigen presentation [69].

### 2.3. Serum Cytokine and Biomarker Levels

Serum cytokine, chemokine and other biomarker levels have been reported in a number of case series. IL-6 and IL-10 were most consistently found to be elevated among patients with ICI-induced myocarditis. Other changes that have been observed include elevation in IL-8, CXCL9, CXCL10, CXCL13, monocyte chemotactic and activating factor (MNCAF), granulocyte-macrophage colony-stimulating factor (GM-CSF), hepatocyte growth factor and vascular endothelial growth factor A (VEGF-A) [97,98,100,101]. Although IL-6, CXCL9, CXCL10 and CXCL13 levels appear to provide modest diagnostic accuracy for identifying patients with ICI-induced myocarditis, these were not fully validated in larger populations, and more data are required before clinical application is feasible [98].

## 3. Animal Models of ICI-Induced Myocarditis

Several mouse models have been described to simulate ICI-induced myocarditis to facilitate research understanding of the pathogenesis of ICI-induced myocarditis and drug discovery. The following section describes these mouse models and the insights they offer regarding the multimodal pathogenesis of ICI-induced myocarditis.

### 3.1. Transgenic Mice

Even before the commencement of the first randomized controlled trial of ICI for malignancies in 2000, it was already known that knocking out genes related to immune checkpoints could lead to myocarditis or dilated cardiomyopathy. Table 3 summarizes transgenic mice models of ICI-induced myocarditis.

#### 3.1.1. CTLA4^−/−^

In 1995, it was found that BALB/c CTLA4^−/−^ mice develop fulminant myocarditis in addition to rapidly lethal lymphoproliferative diseases. Myocardium of the transgenic mice consisted of abundant CD3+ T cells and F4/60+ macrophages [106]. Homozygous knockout of CTLA4 could also be accomplished by cross-breeding CTLA4 floxed C57BL/10.Q mice (CTLA^fl/fl^) and CD4-Cre transgenic mice containing CD4 enhancer, promoter and silencer sequences driving the expression of a Cre recombinase gene (Tg(Cd4-cre)1Cwi). In addition to lymphoproliferative disease and myocarditis, hyperimmunoglobulinemia with elevated levels of IgM, IgG and IgA was observed. The change in IgG level was mainly driven by the elevated IgG1 level [107].

#### 3.1.2. PDCD1^−/−^

In 2001, it was discovered that BALB/c mice with the PDCD1^−/−^ genotype develop dilated cardiomyopathy. Although no infiltration of inflammatory cells was found in the myocardium, linear deposition of IgG, predominantly IgG1, and C3 complements were found on cardiomyocytes [103]. Autoantibodies against cardiac troponin I were later found in serum of transgenic mice, suggesting that homozygous knockout of PDCD1 leads to autoantibody-mediated cardiomyopathy [102]. Subsequently, MRL mice with the PDCD1^−/−^ genotype were shown to develop myocarditis instead of dilated cardiomyopathy [104,105]. Leukocytes infiltrating the myocardium were mainly CD8+ cytotoxic T cells (35.4%), Mac1+Gr1+ myeloid cells (23.6%) and CD4+ helper T cells (19.7%). CD8+ and CD4+ T cells were activated with the expression of CD44, CD69 and CD25. It was determined that the myocarditis was driven mainly by the T_H_1 response as the majority of T cells exhibited the T_H_1 marker Tim3, and mRNA levels of T_H_1 cytokines were elevated with nearly undetectable T_H_2 cytokines [105]. Recently, single-cell RNA and T-cell receptors sequencing analysis revealed the clonally expanded effector memory CD8+ T cells, with increased expression of cytotoxicity marker Nkg7, chemokine marker CCL5 and exhaustion marker Lgals1. Central memory CD4+ T cells and T regulatory cells in the myocardium were increased as well. Furthermore, a high titer of autoantibodies against cardiomyocytes were found in the sera of the transgenic mice. Western blot with cardiomyocytes extract revealed that the autoantibodies recognize a 200 kDa band, although the exact protein involved was not known [99].

#### 3.1.3. PDL1^−/−^

MRL/MpJ-Fas(lpr) PDL1^−/−^ mice were originally bred to investigate lupus nephritis. Unexpectedly, it was found that the transgenic mice died of myocarditis and pneumonitis before developing nephritis. Leukocytes infiltrating the myocardium were predominantly CD68+ macrophages and CD8+ cytotoxic T cells. There were also smaller populations of CD4+ helper T cells and rare B220+ B cells. Most of the infiltrating leukocytes, with the exception of B cells, expressed PD-1. High titers of anti-cardiac myosin autoantibodies were detected in the sera of transgenic mice (1:5300), which were mainly IgG1 and IgG2a, together with a smaller proportion of IgG2b isotypes. Anti-cardiac troponin I was also detected although to a lesser extent (1:733) [105].

#### 3.1.4. PDCD1^−/−^ CTLA4^+/−^

Combinational ICI therapy with anti-PD-1 and anti-CTLA4 monoclonal antibodies is increasing used for treating advanced malignancies [110]. Using C57BL/6J mice with the PDCD1^−/−^ CTLA4^+/−^ genotype, the effect of combinational ICI therapy could be studied. Leukocytic infiltration to the myocardium predominantly consisted of CD8+ cytotoxic T cells and F4/80+ macrophages. There was a relatively low abundance of CD4+ helper T cells and Foxp3+ regulatory T cells. PD-L1 expression was observed in cardiac cells. As circulating microRNA, miR-721 was recently described to be an accurate biomarker for distinguishing myocarditis and other cardiac pathologies; the miR-721 level was also assessed in this ICI-induced myocarditis model. Transgenic mice had higher levels of miR-721 than wild-type strains [108].

### 3.2. Induction with ICI

It has not been possible to induce ICI-induced myocarditis in mice by injecting ICI monotherapy alone. Additional interventions such as combinational ICI, tumor inoculation, cardiac sarcomere sensitization or cardiac irradiation are necessary to induce ICI-induced myocarditis in mice. Table 4 summarizes experimental models of myocarditis induced by ICI.

#### 3.2.1. Combinational ICI

Combinational ICI has been utilized to induce myocarditis in mice. For C57/BL6J wild type mice, combinational anti-PD-1 (25 mg/kg) and anti-CTLA4 (25 mg/kg) given every 3 days for 5 doses led to myocarditis. Leukocytes infiltrating the myocardium were predominantly CD8+ cytotoxic T cells and F4/80+ macrophages. Evidence of cardiomyocyte apoptosis was seen including positive staining on terminal deoxynucleotidyl transferase–mediated deoxyuridine triphosphate nick end labeling (TUNEL) and cleaved caspase 3. RNA sequencing revealed genes involved in reactive oxygen species were increased, while heart specific genes including mesencephalic astrocyte–derived neurotrophic factor (Manf) and heat shock 70-kDa protein 5 (Hspa5) were reduced [108]. Alternatively, ICI-induced myocarditis could also be induced in MRL/MpJ-Fas/lpr mice by giving anti-PD-1 (200 μg) and anti-CTLA4 (200 μg) two times per week for 8 weeks. Subtle leukocyte infiltrate, sarcomere disarray and endothelial cell injury were observed after drug administration [109].

#### 3.2.2. Tumor Inoculation

The motivation to include tumor inoculation in the ICI-induced myocarditis model is to more closely simulate the pathophysiology in cancer patients who use ICIs. In one study, colorectal adenocarcinoma (CT26) was injected via the tail vein of BALB/c mice to induce lung metastasis. Anti-PD-1 (200 μg) and anti-PD-L1 (200 μg) were given after confirmation of lung metastasis establishment. Intriguingly, the infiltrating leukocyte population observed was distinct from other mouse models of ICI-induced myocarditis. Flow cytometry revealed that combinational ICI led to an increase in inflammatory monocytes, while sequential ICI therapy with anti-PD1 followed by anti-PD-L1 resulted in an increase in neutrophils. Unlike other experimental models of ICI-induced myocarditis, no increase in T cells was observed [111]. In another model, C57/BL6J wild type mice were given inoculations of colorectal cancer cells (MC38), melanoma cells (B16F10) or breast cancer cells (EO771), followed by combinational ICI after the tumor size reached 200–250 mm^2^. A combination of anti-PD-1 (25 mg/kg) and anti-CTLA4 (25 mg/kg) given every 3 days for five doses led to lymphocytic myocarditis with predominantly CD8+ cytotoxic T cells in female mice receiving any of the designated cancer cell lines, similar to the pattern observed in other models of ICI-induced myocarditis [108]. Finally, BALB/cByJNarl mice given subcutaneous inoculation of mouse melanoma (B16-F10), followed by anti-PD-1 (250 μg) every 72 h for six doses after the tumor size reached 100 mm^3^ also led to myocarditis. Leukocytes infiltrating the myocardium were predominantly CD4+ helper T cells and CD8+ cytotoxic T cells. Importantly, it was shown that mice given anti-PD-1 without tumor inoculation did not develop myocarditis, suggesting that the presence of an underlying tumor is a contributing factor to myocarditis [112].

#### 3.2.3. Cardiac Sarcomere Immunization

A low level of autoantibodies against cardiac sarcomere may be present even in completely healthy individuals [117]. A higher prevalence and titer of such autoantibodies are present in patients with underlying cardiac diseases such as dilated cardiomyopathy and ischemic heart disease [117,118]. To recapitulate the effect of ICI therapies on patients with preexisting autoimmunity against the myocardium, cardiac sarcomere immunization was performed prior to ICI therapies.

In one report, BALB/c mice were immunized by giving a subcutaneous injection of troponin I peptide (250 μg) together with Freund’s complete adjuvant. Anti-PD-1 (5 mg/kg) was subsequently given. Histological analysis revealed infiltration of inflammatory cells to the myocardium [113,114]. Alternatively, it was demonstrated that BALB/c mice can be immunized by giving a subcutaneous injection of 0.05 mg murine myosin heavy chain α fragment (amino acid 614–629: Acetyl-SLKLMATLFSTYASAD-COOH) together with Freund’s complete adjuvant, and pertussis toxins (500 ng) were given. In the reported experiments, anti-PD-1 (100 μg) was given either concurrently with the myosin heavy chain α fragment or in a delayed fashion. Mice that received myosin heavy chain α fragment immunization with ICI served as a positive control in the experiment. It was found that mice receiving a delayed exposure of ICI had a higher proportion of CD4+ and F4/80+ cells, while mice receiving concurrent ICI had reduced CD4+ cells in the myocardium when compared to the positive control [114]. Finally, another approach of deriving ICI-induced myocarditis is to immunize BALB/c mice using the skeletal muscle homogenate of guinea pigs. In one study, the anti-PD-1 antibody tislelizumab (2 µg/kg) was given to induce myocarditis and myositis. Mice that received an immunization without ICI served as positive controls. Although there was focal inflammatory cells infiltration in the myocardium of positive controls, mice that received ICI had slightly more CD8+ T cells and fewer CD4+ CD25+ Foxp3+ regulatory T cells. Muscle fiber degeneration, necrosis and dissolution were seen in the ICI group. It was also found that autophagy of cardiomyocytes was more severe in the ICI group than positive controls [115].

#### 3.2.4. Cardiac Irradiation

There is insufficient data regarding the safety of concurrently giving thoracic radiotherapy and ICI in clinical practice [119]. There was concern that ICI may exacerbate myocardial inflammation or damage induced by thoracic irradiation. To investigate this clinical question, C57BL/6 mice were given anti-PD-1 (10 mg/kg) for 1 day before receiving thoracic and cardiac irradiation. It was found that mice receiving concurrent ICI had higher mortality at 2 weeks than those that solely received radiotherapy (30% vs. 0%). Furthermore, there were significantly more CD45+ lymphocytes, CD4+ helper T cells, CD8+ cytotoxic T cells and F4/80+ macrophages in the myocardium of the ICI group. Further experiments with CD8+ cells’ depletion with monoclonal antibodies led to reversal of the acute mortality while CD4+ depletion resulted in no significant change. Taken together, these findings suggest that radiotherapy and ICI have additive cardiotoxic effects mainly mediated by CD8+ cytotoxic T cells [116].

### 3.3. Comparison between Animal Models and Human Studies

The current mouse models used to study ICI-induced myocarditis represent an exaggerated and accelerated form of the disease observed in humans, and the pathways of immune system activation seem to be similar in both animal and human models. In both human studies and animal models, the profile of the inflammatory infiltrate in the myocardium is consistently characterized by a T cell infiltrate dominated by CD8+ cytotoxic T cells, with occasional macrophages or monocytes [92,94,95,105,108,109,115,116]. Clonality was observed in some differentially expressed subsets of T cells in the peripheral blood in human studies [99].

In addition, there is a suggestion of activation of the humoral immune system in both human and animal studies. Indirect evidence of complement fixation was noted from human endomyocardial biopsies [92] though disease-causing autoantibodies have yet to be identified. However, autoantibodies directed against cardiac myosin were found in mouse models, especially those of the IgG1 subclass [99,103,105,107]. Since IgG1 is capable of activating the classical pathway of complement, there is reason to suppose that similar antibodies could be identified in humans as well. Identification of such a pathogenic antibody would be of great significance since the current treatment paradigm of ICI-induced myocarditis is targeted at T cell overactivation and does not target the humoral immune system specifically.

### 3.4. Merits and Limitations of Experimental Animal Models

One of the key advantages of transgenic mice is that their myocarditis phenotype is highly reproducible. Furthermore, some of the transgenic strains, such as PDCD1^−/−^ and CTLA4^−/−^ [99,102,103,105,106,107] are available for order from common animal laboratories. Nonetheless, congenital deficiency in immune checkpoint proteins may not necessarily reflect actual pathophysiology of patients with ICI-induced myocarditis, which is acquired inhibition of immune checkpoints in adulthood in the presence of advanced malignancies. In fact, one of the studies demonstrated that knocking out CTLA4 in adulthood with the Cre-Lox recombination system does not lead to myocarditis [107].

There are a wide variety of experimental models of ICI-induced myocarditis that are induced by giving ICIs directly. These models seek to mimic the underlying pathophysiology in human cancer patients. For instance, tumor inoculation allows for studying the interplay between malignancy and the immune system [108,111,112]. Furthermore, the current evidence suggests that a small proportion of individuals have low grade autoimmunity against the myocardium that predisposes them to developing myocarditis after receiving ICIs. The tactic of immunizing mice with cardiac sarcomere before giving ICIs allows researchers to investigate the effect of ICIs in patients with underlying autoimmunity against the myocardium [113,114,115]. It is currently uncertain whether these ICI-induced myocarditis models induced using ICIs are consistently reproducible, as it is likely that the cardiac phenotype derived from these experimental designs are highly dependent on the precise mice strain, ICI dosing regimen, tumor cell line and sarcomere immunization tactic used.

## 4. Drug Screening and Opportunities for Treatment

Myocarditis that occurs as part of systemic autoimmune diseases, such as sarcoidosis, responds well to immunosuppressive therapies. On the other hand, isolated myocarditis of various etiologies has not been shown to benefit from immunomodulatory agents. For instance, use of steroid or intravenous immunoglobulin did not improve the clinical outcome of patients with viral myocarditis [120,121]. Similarly, no immunomodulatory therapies have been convincingly shown to be efficacious for ICI-induced myocarditis. The animal models described herein, which seem to corroborate well with findings from humans, open the door for studies to assess the safety and efficacy of various immunosuppressive agents in the treatment of ICI-induced myocarditis. Among the second line immunomodulatory therapies that have been reported in case series, only abatacept has been tested in animal models [109]. Abatacept is a fusion protein composed of the Fc region of IgG1 fused to the extracellular domain of CTLA4. Abatacept prevents T cell activation by binding to CD80 and CD86 costimulatory molecules [122]. Animal models showed that abatacept increased the rate of myocarditis resolution, reduced the extent of cardiomegaly and more importantly improved survival compared to control transgenic mice [109]. Based on case reports demonstrating successful treatment of ICI-induced myocarditis with abatacept [123], as well as data from animal studies, prospective studies have been designed to investigate its potential in treating this challenging disease entity further. ATRIUM (NCT05335928) and ACHLYS (NCT05195645) are two on-going prospective studies investigating the use of abatacept for treatment of ICI-induced myocarditis.

Despite the widespread use of agents such as mycophenolate mofetil for severe IRAEs including myocarditis [73,124], there is a paucity of high-quality data to support the use of most immunosuppressants in patients with ICI-induced myocarditis. The T cell phenotype described from human and animal models lends a strong biological basis for the use of T cell-directed therapies, including but not limited to anti-thymocyte globulin, calcineurin inhibitors and mycophenolate mofetil. There are no currently available agents that specifically target only CD8+ effector T cells. There is a strong theoretical basis for targeting the three activation signals of T cells described above. T cell activation via signal 1 is targeted by the use of anti-thymocyte globulin as well as agents such as calcineurin inhibitors including tacrolimus or cyclosporin lower downstream. The co-stimulatory signal 2 is targeted by agents such as abatacept; this is conceptually attractive as CTLA4 is the target of ipilimumab and is likely a significant culprit in the pathophysiology of ICI-induced myocarditis [109,123]. Besides mycophenolate mofetil, few agents targeted at cytokines in signal 3 have been studied in the treatment of ICI-induced myocarditis. mTOR inhibitors such as sirolimus and everolimus are attractive agents to consider in this setting, as they target signaling via all of signals 1 to 3 and, unlike other agents, are not oncogenic. In fact, both sirolimus and everolimus have shown significant anti-tumor efficacy, presumably by enriching the population of Foxp3+ regulatory T cells [125,126]; everolimus is already a recognized second-line agent for hormone-positive breast cancer. Agents such as JAK inhibitors, including tofacitinib and baricitinib, target cytokine pathways important in signal 3. The IL-2 pathway is closely associated with the expression of JAK and may be targeted by the monoclonal antibody basiliximab, currently the only anti-IL-2 antibody available commercially [67].

The presence of antibody activation from mouse models and possibly from human studies provides a biological rationale for use of therapies such as rituximab, plasmapheresis or intravenous immunoglobulins [92,99,103,105,107]. That these antibodies may potentially be complement-fixing also suggests that complement-based therapies may be of potential utility, although there is little real-world data to support their use, and the antibody that purportedly leads to ICI-induced myocarditis has yet to be identified.

## 5. Conclusions and Future Directions

The underlying immunopathogenesis of ICI-induced myocarditis is the subject of extensive research using animal models as well as endomyocardial biopsies and serum samples from affected human subjects. The identification of clonal T cell populations that may directly contribute to the pathogenesis of ICI-induced myocarditis, as well as the delineation of the patterns of change in T cell activity and expression in the myocardium and in the peripheral blood, provide further rationale for the adoption of T-cell directed therapies for ICI-induced myocarditis. Given the rarity of ICI-induced myocarditis, large-scale randomized controlled trials will be challenging if not impossible to conduct. It is therefore critical to utilize animal models to screen out promising candidates such that trialists can focus their limited resources into a small number of drugs. Currently, only abatacept has been tested in animal models [109]. The usefulness of other immunomodulatory therapies should be evaluated in animal models before going into human clinical trials.

There are still significant knowledge gaps in the field of ICI-induced myocarditis requiring further research. Chief among these is a lack of biomarkers for predicting patients at risk of developing ICI-induced myocarditis. High-sensitivity troponin assays have been used for identifying patients with early ICI-induced myocarditis [69]. Unfortunately, troponin is elevated in a wide variety of cardiovascular pathologies, including acute coronary syndrome or pulmonary embolism. Experimental models of ICI-induced myocarditis may allow researchers to discover novel biomarkers for early diagnosis of the condition. For instance, circulating Temra cells and miR-721 in PBMCs are promising markers that have emerged from animal studies [99,108].

ICIs have become a cornerstone of contemporary cancer therapy. While there is an increasing number of patients benefiting from these revolutionary treatments, the clinician must be cognizant of the growing wave of patients who will develop IRAEs, including severe and life-threatening complications such as ICI-induced myocarditis. Further research into this emerging clinical entity is urgently warranted, and the results of upcoming trials of immunomodulatory treatments are eagerly awaited.

## Figures and Tables

**Figure 1 biomedicines-11-00107-f001:**
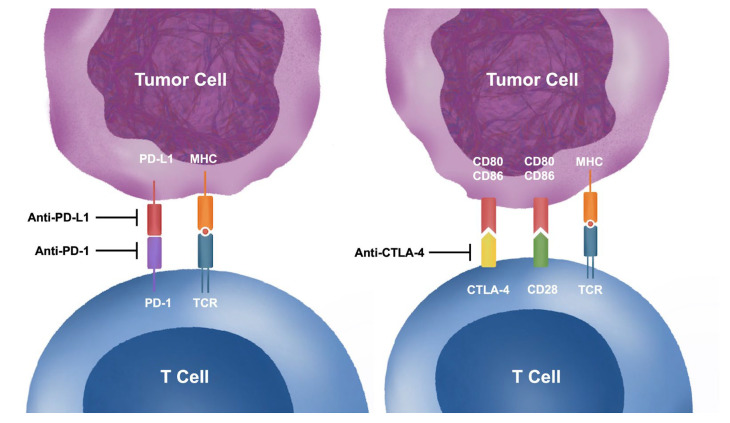
Mechanisms of Immune Checkpoint Inhibitors. Binding of programmed ligand-1 (PD-L1) of tumor cells to programmed death 1 (PD-1) receptors of T cell reduces T cell proliferation, down-regulates cytokine expression and induces apoptosis. Anti-PD-1 and anti-PD-L1 monoclonal antibodies are used to inhibit PD-1/ PD-L1 axis, thereby increasing anti-tumor activities by T cells. CD28 of T cells are co-stimulatory receptors that bind CD80/ CD86 of tumor cells to activate T cells. Cytotoxic T-lymphocyte-associated protein 4 (CTLA-4) competitive bind CD80/CD86 and thereby reduce T cell activation. Anti-CTLA-4 monoclonal antibodies inhibit CTLA-4 activities, increase CD28 co-stimulatory signaling and enhance T cell activities against tumor cells. Abbreviations: CTLA-4, cytotoxic T-lymphocyte-associated protein 4; MHC, major histocompatibility complex; PD-1, programmed death 1; PD-L1, programmed death ligand-1; TCR, T cell receptor.

**Table 1 biomedicines-11-00107-t001:** Manifestation of Immune-Related Adverse Events in Various Organs.

System	Immune-Related Adverse Events
Cardiovascular	Myocarditis and heart failure, pericarditis and pericardial effusion, arrhythmia, sudden cardiac death, hypertension [41,42]
Respiratory	Pneumonitis, sarcoidosis [43,44,45,46]
Neurological	Encephalitis and encephalopathy, meningitis, transverse myelitis, Guillain Barré syndrome, posterior reversible encephalopathy syndrome, multiple sclerosis, neuropathy, myasthenia gravis [47,48,49]
Renal	Glomerulonephritis (including nephrotic syndrome), interstitial nephritis, acute tubular necrosis, renal failure [50,51,52]
Gastrointestinal	Gastritis, enteritis, colitis, gastroenteritis, hepatitis, pancreatitis [53,54]
Endocrine	Thyroiditis, autoimmune hyperthyroidism and hypothyroidism, hypophysitis, adrenalitis and primary adrenal insufficiency, autoimmune diabetes [55]
Hematological	Cytopenias (commonly thrombocytopenia and leucopenia), hemophagocytic lymphohistiocytosis, aplastic anaemia, hemolytic anemia, acquired hemophilia and other coagulopathies [56,57]
Rheumatic/Musculoskeletal	Arthritis, myositis, fasciitis, vasculitis, polymyalgia-like syndrome, dermatomyositis, sicca syndrome [58]
Skin	Morbilliform exanthem, lichenoid reactions, vitiligo-like depigmentation, psoriasis, alopecia areata, bullous pemphigoid, Stevens-Johnson syndrome/toxic epidermal necrolysis, Drug Reaction with Eosinophilia and Systemic Symptoms (DRESS) [59,60]
Eye	Episcleritis, conjunctivitis, uveitis, retinopathy, orbital inflammation [61]

**Table 2 biomedicines-11-00107-t002:** Human Studies of Immune Checkpoint Inhibitor-induced Myocarditis.

Specimen	Findings
Endomyocardial biopsy	Leukocyte infiltration predominated by CD8+cytotoxic T cells, with smaller proportion of CD68+ or CD163+ macrophages and CD4+ helper T cells [92,93,94,95]. PD-L1+ staining of cardiomyocytes adjacent to site of injury and infiltrating macrophages were reported [92,94,95]. C4d staining was observed in necrotic cardiomyocyte in one study, suggesting role of antigen-antibody interaction and immune complex formation with complement fixation [92]. Transcriptomic studies demonstrated increased expression of genes involving in multiple inflammatory pathways, especially interferon responses [95].
Peripheral blood mononuclear cells	Neutrophil-to-lymphocyte ratio was increased in ICI-induced myocarditis patients [97]. Increased T_H_1, T_H_17 and regulatory T cells; decreased T_H_2 cells [98]. Smaller proportion of naïve CD8+ cytotoxic T cells and larger proportion of terminally differentiated effector memory CD45RA re-expression CD8+ T cells (Temra) [99].
Plasma	Consistently elevated in multiple studies: IL-6 and IL-10. Other cytokines that were reported to have increased levels: IL-8, CXCL9, CXCL10, CXCL13, MNCAF, GM-CSF, hepatocyte growth factor and VEGF-A [97,98,100,101].

**Table 3 biomedicines-11-00107-t003:** Transgenic Mice for Immune Checkpoint Inhibitor-induced Myocarditis.

Mice Strain	Genotype	Cardiac Phenotype
BALB/c	PDCD1^−/−^	DCMP [102,103]
MRL/Mpj	PDCD1^−/−^	Myocarditis [99,104]
MRL/MpJ-	PDL1^−/−^, Fas(lpr)	Myocarditis [105]
BALB/c	CTLA4^−/−^	Myocarditis [106]
Cross breed C57BL/10.Q and Tg(Cd4-cre)1Cwi	CTLA4^fl/fl^, CD4-Cre	Myocarditis [107]
C57BL/6J	CTLA4^+/−^, PDCD1^−/−^	Myocarditis [108,109]

DCMP: dilated cardiomyopathy.

**Table 4 biomedicines-11-00107-t004:** Immune Checkpoint Inhibitor Monoclonal Antibodies for Myocarditis Induction.

Mice Strain	Monoclonal Antibodies Therapy	Tumor Inoculation	Cardiac Sarcomere Immunization	Additional Therapy	Cardiac Phenotype
MRL/MpJ-Fas(lpr)	Combinational anti-PD-1 (200 μg) and anti-CTLA4 (200 μg) 2 times per week for 8 weeks	None	None	None	Myocarditis [109]
C57/BL7J	Combinational anti-PD-1 (25 mg/kg) and anti-CTLA4 (25 mg/kg) every 3 days for x days	None	None	None	Myocarditis [108]
C57/BL7J	Combinational anti-PD-1 (25 mg/kg) and anti-CTLA4 (25 mg/kg) every 3 days for x days after tumor size reaches 200–250 mm^3^	Colorectal cancer (MC38), melanoma (B16F10) and breast cancer (EO771)	None	None	Myocarditis [108]
BALB/c	Combinational anti-PD-1 (200 μg) and PD-L1 (200 μg) on days 0, 2 and 4 after confirmation of lung metastasis	Colorectal adenocarcinoma (CT26) tail vein injection to induce lung metastasis	None	None	Myocarditis [111]
BALB/c	Sequential anti-PD-1 (200 μg) on days 0, 2 and 4, followed by anti-PD-L1 (200 μg) on days 6, 8 and 10 after confirmation of lung metastasis	Colorectal adenocarcinoma (CT26) tail vein injection to induce lung metastasis	None	None	Myocarditis [111]
BALB/c	Sequential anti-PD-L1 (200 μg) on days 0, 2 and 4, followed by anti-PD-1 (200 μg) on days 6, 8 and 10 after confirmation of lung metastasis	Colorectal adenocarcinoma (CT26) tail vein injection to induce lung metastasis	None	None	Myocarditis [111]
BALB/cByJNarl	Anti-PD-1 (250 μg) every 72 h for 6 doses after tumor size reaches 100 mm^3^	Mouse melanoma cells (B16-F10) subcutaneous injection	None	None	Myocarditis [112]
BALB/c	Anti-PD-1 (5 mg/kg) on days 7, 9, 11, 13 and 15	None	Murine troponin I peptide (250 μg) subcutaneous injection on days 0 and 7	Freund’s complete adjuvant on days 0 and 7	Myocarditis [113]
BALB/c	Anti-PD-1 (0.1 mg) on days 14, 16, 8 and 20	None	Murine myosin heavy chain α (MHCα) fragment (amino acid 614–629: Acetyl-SLKLMATLFSTYASAD-COOH) subcutaneous injection on days 0 and 7	Freund’s complete adjuvant on days 0 and 7; and Pertussis toxin (500 ng) on day 0	Myocarditis [114]
BALB/c	Anti-PD-1 (0.1 mg) on days 0, 2, 4 and 6	None	Murine myosin heavy chain α (MHCα) fragment (amino acid 614-629: Acetyl-SLKLMATLFSTYASAD-COOH) subcutaneous injection on days 0 and 7	Freund’s complete adjuvant on days 0 and 7; and Pertussis toxin (500 ng) on day 0	Myocarditis [114]
BALB/c	Anti-PD-1 (2 μg/kg) on weeks 5 and 6	None	Skeletal muscle homogenate of guinea pigs (0.25 mL) once per week for 6 weeks	Freund’s complete adjuvant (0.25 mL) once per week for 6 weeks	Myocarditis [115]
C57BL/6	Anti-PD-1 (10 mg/kg) 1 day before radiotherapy	None	None	Cardiac irradiation	Myocarditis [116]

## Data Availability

Not applicable.

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
