# Peer review of "Immunopathogenesis of Immune Checkpoint Inhibitor Induced Myocarditis: Insights from Experimental Models and Treatment Implications"

_biomedicines, 2023, doi:10.3390/biomedicines11010107_

Round 1

Reviewer 1 Report

The review describes the myocarditis induced by ICIs comprehensibly and is informative. However, there are few concerns. Long paragraphs and sentences have been used In long paragraphs only one to two citations have been used. Experimental strategies discussed in original articles have been discussed again- this can be omitted.

1.       Lines 38-44- too long sentence, please split.

2.       ICIs function by modulating the altered immune response to proliferation of cancer 49 cells.- not a complete sentence

3.       Please use a figure to depict the mechanism of action of ICIs targeting ligands- section 1.2

4.       Lines 111-116- very long sentence

5.       Lines 116-120- please cite

6.       Please include a table including organ-related symptom in section 1.3.1

7.       Section 2.1 is from four article 60, 61, 62, and 63.

8.       However, many of the above studies were uncontrolled case series- before this there is only one citation, please mention bout which studies the authors are talking about

9.       Lines 242-259 are from one article? Please cite more articles

10.   Section 2.1 to 2.3 can be summarized in a table describing all biomarkers.

11.   Section 3.2.2- large text is supported by only one citation each?

12.   Lines 406-426- the text should be discussed in short; citation is sufficient for the readers to go through the experimental strategy. Please discuss in a concise manner

13.   Same issue with lines 501 to 520

Author Response

Response to Reviewer 1

  1. Lines 38-44- too long sentence, please split.

Response: Thank you for the advice. The original sentence was separated for easier comprehension.

  1. ICIs function by modulating the altered immune response to proliferation of cancer 49 cells.- not a complete sentence

Response: Thank you for providing us with the feedback. The sentence was rewritten to improve clarity.  

  1. Please use a figure to depict the mechanism of action of ICIs targeting ligands- section 1.2

Response: Thank you for the suggestion. Figure 1 was added to illustrate mechanisms of ICIs.

  1. Lines 111-116- very long sentence

Response: Thank you for the feedback. The sentence was separated to enhance comprehensibility.

  1. Lines 116-120- please cite

Response: Thank you for the advice. Citation to two relevant papers were added.

  1. Please include a table including organ-related symptom in section 1.3.1

Response: Thank you for the advice. Table 1 was added to summarize types of IRAE that can occur and their corresponding manifestations.

  1. Section 2.1 is from four article 60, 61, 62, and 63. However, many of the above studies were uncontrolled case series- before this there is only one citation, please mention bout which studies the authors are talking about

Response: Thank you for the suggestion. To improve clarity, we cited the relevant articles early in section 2.1. to better orient readers. Also explained that most studies  published to date are case series without case-control comparisons. The data provided by these reports are nevertheless invaluable owing to the rarity of the disease entity.

  1. Lines 242-259 are from one article? Please cite more articles

Response: Transcriptomic study of endomyocardial tissue allows researchers to better understand pathogenic mechanisms and consequences of ICI myocarditis. Unfortunately, to date RNA sequencing of endomyocardial tissue from ICI-induced myocarditis patients has only been reported in 1 peer reviewed article. The relevant section was updated to emphasize the rarity of such valuable reports.

  1. Section 2.1 to 2.3 can be summarized in a table describing all biomarkers.

Response: Thank you for the suggestion. Table 2 was added to the latest manuscript to summarize findings from 2.1. to 2.3. 

  1. Section 3.2.2- large text is supported by only one citation each?

Response: Thank you for the enquiry. To the best of our knowledge, only three papers reported animal models of ICI-induced myocarditis which involved inoculation of tumor cell lines. The relevant section was rewritten to more succinctly summarize methodology and findings from the papers.

  1. Lines 406-426- the text should be discussed in short; citation is sufficient for the readers to go through the experimental strategy. Please discuss in a concise manner 

Response: Thank you for the important feedback. The relevant section was simplified to enhance comprehensibility.

  1. Same issue with lines 501 to 520

Response: Thank you for the advice. The section about cardiac irradiation experiments was simplified to improve readability.  

Reviewer 2 Report

Wong and co-workers present a detailed review article on the current literature on the pathogenesis of myocarditis induced by immunotherapy (checkpoint inhibitors).

Comments:

The article is very detailed, sometimes too detailed in some places regarding the experimental conditions in the individual studies. These details read complicated and would not necessarily be needed for understanding. Shortenings could be made here without losing important information for the readers.

What I miss, however, is a section that goes at least a little bit into the pathogenesis of myocarditis in general, different triggers, clinical pictures, therapy, etc.

I suggest summarizing the most important points of the article graphically. This would certainly be helpful for the readers.

Minor comments:

- Line 166: One including should be deleted

- Line 320: A band of 200 kDa represents a protein rather than a peptide.

- Table 2: The columns of the table are partly very close to each other and therefore difficult to read. This should be revised.

- The text has unnecessary spaces in some places.

Reviewer 3 Report

In this paper, the authors performed a detailed review of a current topic such as ICIs myocarditis, evaluating the pathogenetic mechanisms and therapeutic options by comparing the available human data with data from animal trials. I believe it is a comprehensive, well-written work of absolute topical interest.

Round 2

Reviewer 1 Report

No more concerns